# Eosinophil Cationic Protein Variation in Patients with Asthma and CRSwNP Treated with Dupilumab

**DOI:** 10.3390/life13091884

**Published:** 2023-09-08

**Authors:** Andrea Giovanni Ledda, Giulia Costanzo, Giada Sambugaro, Cristiano Caruso, Martina Bullita, Maria Luisa Di Martino, Paolo Serra, Davide Firinu, Stefano Del Giacco

**Affiliations:** 1Department of Medical Sciences and Public Health, University of Cagliari, 09100 Cagliari, Italy; andrea.giovanni.ledda@gmail.com (A.G.L.); giuliacostanzo14@gmail.com (G.C.); giadasambugaro@gmail.com (G.S.); mdimartino@aoucagliari.it (M.L.D.M.); paoloserra@cagliarirespira.it (P.S.); delgiac@gmail.com (S.D.G.); 2UOSD DH Medicina Interna e Malattie dell’Apparato Digerente, Fondazione Policlinico A. Gemelli IRCCS, 20123 Rome, Italy; cristiano.caruso@policlinicogemelli.it; 3Department of Medical Sciences and Public Health, Faculty of Medicine, Università Cattolica del Sacro Cuore, 00168 Rome, Italy; 4Faculty of Medicine and Surgery, University of Cagliari, 09100 Cagliari, Italy; martinabullita96@gmail.com

**Keywords:** asthma, nasal polyps, eosinophils cationic protein, dupilumab, biomarkers, type 2 inflammation

## Abstract

Background: Asthma is a clinical syndrome characterized by recurrent episodes of airway obstruction, bronchial hyperresponsiveness and airway inflammation. Most patients with asthma present a “type 2” (TH2) inflammation. ILC2 and TH2 cells release cytokines IL4, IL-13 and IL-5. CRSwNP is a condition characterized by hyposmia or anosmia, nasal congestion, nasal discharge, and face pain or pressure that last for at least 12 weeks in a row without relief. Both asthma and CRSwNP are often characterized by a type 2 inflammation endotype and are often present in the same patient. Dupilumab is a fully human monoclonal antibody targeting the interleukin-4 receptor α (IL-4Rα) subunit, blocking IL4/IL-4Rα binding and IL13. It has been labelled for the treatment of moderate to severe asthma in patients from the age of 12 years with an eosinophilic phenotype, and it has demonstrated efficacy and acceptable safety. Our study aims to investigate the effects of dupilumab on type 2 inflammatory biomarkers, such as eosinophils and eosinophil cationic protein (ECP). ECP is an eosinophil-derived substance contained in granules that are released during inflammation and causes various biological effects, including tissue damage in asthmatic airways. Methods: ECP, Eosinophil counts (EOS), and total immunoglobulin E (IgE) levels were longitudinally measured using immunoassays in the serum of 21 patients affected by CRSwNP, of which 17 had asthma as a comorbidity, receiving 300 mg dupilumab every two weeks. Results: The EOS and ECP, after a first phase of significant increase due to the intrinsic characteristic of the block of IL-4 and IL-13, returned to the baseline 10 months after the initial administration of dupilumab. Fractional exhaled nitric oxide (FeNO) and serum total IgE decreased significantly after 9 months. Asthma Control Test (ACT) scores improved after dupilumab treatment. FEV1% and FEV1 absolute registered a significant improvement at 10 months. Conclusions: Patients who received 300 milligrams of dupilumab every two weeks first experienced a temporary increase in eosinophils (EOS) and eosinophil cationic protein (ECP), then exhibited a gradual decline in these variables with a subsequent return to the initial baseline levels. When compared to the baseline, we observed that the levels of IgE and FeNO decreased over time, while there was an increase in both FEV1 and FEV1%.

## 1. Introduction

Asthma is a clinical syndrome characterized by recurrent episodes of airway obstruction, bronchial hyperresponsiveness and airway inflammation. Among the most frequent symptoms are shortness of breath, cough, wheezing, and chest tightness [1]. Asthma is a chronic respiratory disease characterized by phenotypic heterogeneity. It currently impacts over 300 million individuals globally. There exist several distinct asthma phenotypes, each of which experiences substantial influence from a varied array of cytokines, chemokines, and other proinflammatory mediators generated by immune–inflammatory cells and structural cells present in the airways. Eosinophils are known to have a pivotal role in the pathophysiology of asthma, serving as the primary inflammatory cells implicated in the disease process. The processes involved in the maturation, activation, survival, and recruitment of eosinophils in the bronchial wall and airway lumen are of significant importance in understanding the development of both allergic and nonallergic asthmatic phenotypes [2]. It is well known that asthma is a pathology that can present in a broad variety of ways and that treatment results can vary significantly from patient to patient [3]. Most asthmatics patients present type 2 inflammation that is mainly mediated by mast cells, eosinophils, IgE, T helper cells 2 (TH2 cells) and basophils. Type 2 innate lymphoid cells (ILC2) and TH2 cells stimulate the release of Interleukin 4 (IL-4), interleukin 13 (IL-13) and Interleukin 5 (IL-5) [1].

Chronic rhinosinusitis with nasal polyps (CRSwNP) is a medical condition characterized by a significant decline in quality of life, impacting approximately ten percent of the adult population in the United States and Europe. CRSwNP is an abbreviation denoting the medical condition known as chronic rhinosinusitis with nasal polyps. This condition is defined by the manifestation of symptoms such as hyposmia or anosmia, nasal congestion, nasal discharge, and facial pain or pressure that persist for a minimum duration of 12 consecutive weeks without relief. Furthermore, this disorder is distinguished by the manifestation of symptoms such as anosmia or hyposmia [4]. Individuals diagnosed with chronic rhinosinusitis with nasal polyps (CRSwNP) may present with elevated concentrations of eosinophil cationic protein (ECP), which serves as an indicator of active eosinophils that have undergone degranulation. Furthermore, these patients demonstrate heightened levels of eotaxins and total immunoglobulin E (IgE), as well as interleukin (IL)4, IL5, and IL13 in their blood, plasma, and nasal polyp tissue. Type 2 Innate Lymphoid Cells (ILC2s) and type 2/Th2 lymphocytes have been observed to accumulate in the nasal polyps and mucosa of patients with chronic rhinosinusitis with nasal polyps (CRSwNP). These cells are known to secrete interleukin-4 (IL-4), interleukin-5 (IL-5), and interleukin-13 (IL-13), which play crucial roles in driving type 2 inflammation [5].

The JESREC study’s primary objective is to develop a set of clinical diagnostic criteria for the condition known as eosinophilic chronic rhinosinusitis (ECRS). In this study, a scoring system is developed that takes into consideration a variety of parameters, including the affected side of the disease, the existence of nasal polyps, the findings from facial CT scans, and the peripheral eosinophil count [6]. In addition to this, an algorithm was created that takes into account both the JESREC score and the existence of comorbidities to assess prognosis or the risk of recurrence following surgery on the basis of the endotype (e.g., eosinophilic endotype is connected with the major risk of recurrence of nasal polyps after surgery). In the end, this scoring system and algorithm serve as helpful tools that assist clinicians in picking the treatment that is most appropriate for each individual patient’s particular condition [6].

Based on the results of two studies involving Japanese individuals, it was observed that the presence of comorbid asthma was commonly associated with eosinophilic chronic rhinosinusitis (CRS), regardless of the presence of polyps [7,8].

Both asthma and CRSwNP are marked by a defect of the epithelial barrier, and they frequently have the same type 2 immunopathogenesis. It has been demonstrated that CRSwNP patients who also have asthma have an upregulation of type 2 cytokines as well as an IgE-mediated release of immune mediators in both the upper and lower airways. Patients who have type 2 CRSwNP typically have a more severe form of the disease, characterized by a high polyp recurrence rate and asthma that is difficult to treat. CRSwNP that is associated with asthma is characterized by eosinophilia and a high local IgE level [9]. Recent advances in our understanding of the pathophysiologic processes that underlie asthma and nasal polyposis highlight the therapeutic efficacy of targeting eosinophils, type 2 cytokines, and their receptors [10].

Dupilumab is a monoclonal antibody that targets the interleukin-4 receptor (IL-4R) subunit specifically. As a result of its action, it has the effect of inhibiting the binding of IL4/IL-4R and IL-13. This drug has been approved for the treatment of severe asthma in individuals who are at least 12 years old, and who have an eosinophilic phenotype or are in need of oral corticosteroid therapy. Individuals must meet both of these criteria in order to take the medication. In pivotal trials, the efficacy of Dupilumab in mitigating severe asthma exacerbations and boosting lung function was proven through the measurement of spirometric outcomes (particularly FVC and FEV1), as well as assessments of asthma control and quality of life scores. This indicated that dupilumab is effective in mitigating severe asthma exacerbations and enhancing lung function [11]. The efficacy and safety of dupilumab in patients with CRSwNP were shown by a phase 2 randomized double-blind trial controlled by a placebo [12]. The purpose of this study was to explore the effects of dupilumab on individuals who suffered from CRSwNP, whether or not they also suffered from asthma. In comparison to the group that was given a placebo, the group that was given dupilumab had significant improvements in endoscopic, radiographic, and quality of life (QoL) measures. The observed clinical changes were followed by a statistically significant drop in the levels of type 2 biomarkers that were measured in the peripheral blood, specifically total IgE and eotaxin-3. Certain patients experienced transient increases in their blood eosinophil count after the beginning of treatment with dupilumab [5]. The inhibition of the interleukins IL-4 and IL-13 involved in the inflammatory process has, therefore, proved to be an effective therapeutic strategy to keep the three phenotypes of severe asthma under control, as well as the related type 2 pathologies, which often manifest as comorbidities in the patient with severe asthma, aggravating the clinical course [13].

Eosinophil cationic protein (ECP) is a granule-bound eosinophil-derived peptide that damages asthmatic airway tissue when released during inflammation [14]. ECP is higher in symptomatic asthmatic patients than in asymptomatic asthmatic patients [11].

The serum levels of ECP have been linked to the degree of eosinophilic inflammation, suggesting that it could be an indirect marker of eosinophilic activity [15]. The levels of eosinophil cationic protein (ECP) were also deemed to be a reliable prognostic indicator for the recurrence of nasal polyps following surgical intervention. Based on this theoretical framework, it is postulated that a patient diagnosed with chronic rhinosinusitis with nasal polyps (CRSwNP) who exhibits a preoperative eosinophil cationic protein (ECP) level below a specified threshold of 21.8 U/mL is highly likely, with a 95% probability, to achieve a satisfactory surgical outcome, provided that there is no occurrence of early polyp recurrence [16]. In the literature, some studies have analyzed the course of ECP during therapy with anti-IL-5/5R mAbs approved for severe asthma (e.g., mepolizumab, benralizumab) [11,17]. A small study showed high levels of both ECP and eosinophil peroxidase as predictive of the future development of asthma among those with allergic rhinitis [18].

The utility of any assay that quantifies eosinophils and/or assesses eosinophil activation is entirely contingent upon the specificity of the biomarker for eosinophils [19]. Despite extensive research on the temporal pattern of eosinophil count following dupilumab treatment, there is a paucity of knowledge regarding the dynamics of eosinophil-derived chemicals, such as eosinophil cationic protein (ECP). We analyzed the time course of serum EOS, ECP total IgE, FeNO and lung function parameters in patients with severe asthma treated with dupilumab. We also administered and analyzed the ACT (Asthma Control Test), a brief patient-centered assessment tool that can be used to evaluate the amount of asthma control, either in conjunction with or independently of lung function tests [20] and SNOT22 (Sino-Nasal Outcome Test 22)—a useful tool in evaluating the effects of CRS on the quality of life of patients, as well as for researching outcomes in the field of rhinology [21]—scores at 3 months and 10 months compared with the baseline.

The observational study employed a real-world methodology, wherein physicians faced the proven efficacy of the treatment derived from randomized control trials with the complex factors present in the real-life setting, in order to provide data on the treatment effectiveness and safety [22].

## 2. Materials and Methods

### 2.1. Study Design and Patient Population

This prospective real-life observational study enrolled patients diagnosed with asthma grades 3-4-5 according to GINA [23], and CRSwNP, who were treated with dupilumab. All patients received their diagnosis of asthma according to the ERS guidelines [24], and were referred to the Allergology and Clinical Immunology Unit of the Teaching Hospital, University of Cagliari. For a general comparison of baseline ECP levels, we sampled a group of patients with moderate asthma (GINA steps 3-4), not on biologicals. The study was performed according to the Helsinki Declaration, obtaining valid written consent by participants (SANI study approved on 23-07-2018; #15; MANI study approved on 05-10-2022 #5 by AOU ethical committee). Patients with past smoking history were included. We enrolled a total of 21 patients treated with dupilumab and 21 asthmatic patients not on biologicals who represented the control group (Table 1). Chronic rhinosinusitis with nasal polyps (CRSwNP) was observed in all patients, regardless of whether they were administered dupilumab or assigned to the control group [6]. Individuals belonging to the dupilumab group exhibited moderate to severe CRSwNP, while those in the control group presented a milder manifestation of chronic rhinosinusitis (CRS). A total of 17 individuals in the dupilumab group were asthmatic (85%), whereas every participant in the control group presented with asthma. A total of 9 patients in the group treated with dupilumab and 11 in the control group were atopic (comorbid with an allergic disease like Allergic Asthma or Allergic Rhinocongiuntivitis or Atopic Dermatitis).

### 2.2. Study Design and Measurements

Dupilumab was administered every 2 weeks for 54 weeks, and the levels of various biomarkers were serially analyzed. We scheduled a baseline Visit V0 and a follow-up visit at 3 and 10 months after baseline. At the baseline visit, for all the patients, study and control group, we collected information about bio-anthropometric features, comorbidities and medications. At the baseline and each other visit, patients were assessed for their Asthma Control Test (ACT) score [20], Sino-Nasal Outcome Test on 22 Items (SNOT 22) [21], and complete blood count and ECP levels. ECP was assessed using automated Immunocap TM technology and Phadia 250 system (ThermoFhiser scientific, Uppsala, Sweden) with serum that was obtained from blood samples that had been centrifuged after being left at 25 °C for 60 min, by REMI ELEKTROTECHNIK LIMITED XS R-5S+ (VASAI-401 208, India). The detectable range of ECP is 2–200 U/mL; in healthy adults, the normal ECP serum level is below 14.9 U/mL.

Eosinophil count was assessed using a whole-blood sample collected in a specimen containing Ethylenediaminetetraacetic acid (EDTA), then analyzed using Fluorescence Flow Cytometry with a DASITXN 1500 (Sysmex corporation, Kobe, Japan)

Total IgE was assessed using ImmunoCAP and Phadia 250 system (Thermofisher scientific AB, Uppsala, Sweden), a fully quantitative assay that determines circulating IgE levels over a measurement range from 2 to 5000 kU/L.

Pulmonary function testing was obtained using a Vyaire Medical Vyntus™ BODY Plethysmograph (CareFusion, Hoechberg, Germany); this, and the measurement of fractional exhaled nitric oxide (FeNO), using NIOX VERO (Circassia AB, Uppsala, Sweden), were performed before starting dupilumab (T0, baseline visit) and after 3 (T3m) and 10 (T10m) months. The first endpoint was to describe the course of ECP and eosinophils at 3 and 10 months after the first dupilumab dose and compare it with the baseline.

### 2.3. Statistical Analysis

A Bartlett test was carried out in order to evaluate the differences that were observed between the samples. The Mann–Whitney U test was applied in order to investigate the disparities that could be observed between the two groups. The Spearman correlation coefficient was applied so that we could study the link that existed between the biomarkers. Following the completion of a two-way analysis of variance, the Dunnett multiple comparisons test was used to the clinical index and biomarkers at each time point.

## 3. Results

We included adult patients with eosinophilic asthma of grades 3-4-5 (according to GINA) and CRSwNP, who had previously undergone OCS or surgery. A total of 21 patients initiated dupilumab treatment: 11 were female and 10 were male, and the median age was 45 y (IQR 27.3) (Table 1).

All patients were affected by CRSwNP and asthma, 11 were atopic, and 10 were non-atopic. At baseline, the ACT median score of those with asthma was 21 (IQR 6.4), the median FeNO was 48 ppb (IQR 32.3), the median IgE was 170 (IQR 353), the median EOS was 500 (IQR 366.7) and the median ECP was 120 (IQR 128). Dupilumab was administered every 2 weeks by subcutaneous injections; two patients had an indication for asthma, so they were administered an initial dose of 600 mg once.

At baseline, the level of serum ECP in the group of patients that later started dupilumab was higher but not significantly different than that of the asthmatic (GINA 3-4) group (*p* = 0.35) (Table 2).

During dupilumab, the circulating eosinophil counts started from a baseline median of 450 cell/mm^3^ (IQR 233) and then significantly increased at 3 months up to a median of 860 cell/mm^3^ (IQR 1079) (*p* = 0.013), and later returned to a median of 300 cell/mm^3^ (IQR 833) at 10 months (*p* = 0.93 vs. baseline) (Figure 1a) (Table 3). The levels of ECP also significantly increased from a baseline median of 69.5 U/mL (IQR 65) to a median of 107 at 3 months (IQR 118) (*p* = 0.03), and at 10 months we observed a reduction versus baseline to a median of 36.5 U/mL (IQR 155) (*p* = 0.90) (Figure 1b). Overall, we observed that 52% of patients in our cohort had a rise in peripheral blood eosinophil counts compared with the baseline, 45% of whom reached an eosinophil count above 1500 cells/mm^3^ during the 10 months of observation. No patient showed an eosinophil count above 3200 cells/mm^3^ during the observation period. Regarding ECP levels, 43.5% of patients had a rise in ECP during the study. A total of 33% of patients had a rise in both ECP and eosinophil counts. Besides that, the results of our research showed that there was a correlation between the eosinophils and ECP readings at each time point (Appendix A).

No patient showed symptoms or clinical events that were attributable to hypereosinophilia.

We also evaluated spirometry, FeNO, serum IgE, the Asthma Control Test (ACT) questionnaire and the Sino-Nasal Outcome Test 22 (SNOT22) (Table 3). At 10 months, the absolute FEV1 improved significantly with respect to the baseline of 130 mL (*p* = 0.05) (Figure 1c). Similarly, the FEV1% significantly improved versus baseline in the overall analysis from a median of 96.5% (IQR 28) to a median of 104% (IQR 16) (*p* = 0.01) at T3m, and to a median of 104 (IQR 16) at T10m (*p* = 0.03) (Figure 1d). FeNO and total serum IgE showed a significant reduction over time as well. Total IgE decreased at 3 months from a baseline of 247 UI/mL (IQR 333) to a level of 99 UI/mL (IQR 191) (*p* = 0.002), and the reduction was maintained at 10 months (*p* = 0.01). Similarly, the FeNO decreased significantly at 3 months (*p* = 0.04) and the reduction was maintained at 10 months (*p* = 0.015). SNOT-22 greatly improved after 3 months from a baseline of 66.5 (IQR 34) to 42 (IQR 46.2) (*p* = 0.01), and then the improvement continued at 10 months to 27 (IQR 44) (*p* = 0.02) (Figure 2). The gain in ACT score was not statistically significant (*p* = 0.09) between T0 and T10m, mainly because most patients were non-severe asthmatics with CRSwNP and often had a normal baseline ACT (Table 3). As a relationship between the baseline ECP level and an improvement in FEV1 has been observed in other studies [25], we further investigated if the change in ECP values at 3 months and 10 months of treatment by dupilumab played a role in the improvement in FEV1 toward baseline. Our results showed that there was no significant difference when comparing the improvement in FEV1 at T10m vs. the T0 of those with an elevated baseline ECP (>14 U/mL), compared with the subgroup with a normal ECP (*p*-value = 0.66). Similarly, we did not observe significant changes for FEV1, FEV1%, FeNO, total IgE or SNOT22 according to the baseline EOS or ECP value in our study.

## 4. Discussion

In patients with allergic rhinitis or allergic asthma, the eosinophil mediators have the potential to induce inflammation in the adjacent tissues, specifically the nasal mucosa and the tracheal/bronchial mucosa. In individuals who have been diagnosed with chronic rhinosinusitis (CRS), it has been observed that specific procedures can potentially induce eosinophilic infiltration. The measurement of ECP has also indicated its potential as a biomarker for the diagnostic purposes of Chronic Eosinophilic Rhinosinusitis (ECRS), as it is thought more accurately reflect active eosinophilic inflammation compared to alternative biomarkers [18]. Furthermore, the upregulation of ECP is greater in CRSwNP patients than it is in CRSsNP patients [26]. A study suggest that the production of eosinophil cationic protein (ECP) is facilitated by neutrophils via an immunoglobulin E (IgE)-dependent mechanism. Nevertheless, there is currently a lack of documented cases regarding the IgE-mediated generation of eosinophil cationic protein (ECP) in individuals suffering from allergic asthma. Furthermore, it has been noted that the administration of anti-interleukin-4 (IL-4) medication does not result in a reduction in eosinophil cationic protein (ECP) levels in the serum of these individuals [27]. Another study shows how in pediatric patients who have been diagnosed with asthma, the value of plasma eosinophil cationic protein (ECP) concentrations is a possible biomarker for measuring Type 2 inflammation, corticosteroid therapy response, and exacerbation outcomes. These findings provide greater insight regarding the importance and assessment of eosinophil activation in pediatric patients who suffer from asthma. The answer to the question of whether or not adolescents with elevated plasma ECP concentrations display distinct reactions to alternate management techniques for Type 2 inflammation, such as biologics, is of yet unknown [28].

We assessed how the ECP levels and the peripheral blood eosinophil counts changed over time during the first course of dupilumab in CRSwNP asthmatic patients. We first analyzed the precise time course of ECP levels in the setting of a real-world observational study. Our analysis showed that at the third month of treatment with dupilumab, there is a significant transient increase in EOS counts in 52% of patients and of ECP levels in 43.5% of patients, which is consistent with a previous report of an RCT [29]. The observed increase in eosinophil count in the peripheral blood fits with the hypothesis that the administration of dupilumab hinders the activities of interleukin-4 and interleukin-13 in promoting eosinophil survival, activation, and migration to tissues. However, it does not impact the release of eosinophils from the bone marrow, a process regulated by interleukin-5 [29]. The said increase is followed by a decline towards levels slightly below baseline at 10 months. These kinetics are in contrast with what happens during IL-5 targeted therapy with mepolizumab or benralizumab: during therapy with one of these mAbs, the eosinophil count and the ECP value decrease constantly [11,17].

In contrast to IL-5, IL-4 and IL-13 have a relatively minor direct impact on the number of eosinophils that are produced by the bone marrow. In contrast, the principal effect that is anticipated to result from the administration of dupilumab, which works by inhibiting the IL-4 and IL-13 pathways, is the disruption of the migration of eosinophils, which makes it possible for the number of circulating cells to increase [30]. To promote eosinophil adherence and migration, IL-13 enhances the expression of vascular cell adhesion molecule 1 on the vascular endothelium. Circulating eosinophils rise when dupilumab inhibits IL-13 because, presumably, cell migration is inhibited. Additionally, IL-13 controls the expression of eotaxin as well as thymus- and activation-regulated chemokines. When IL-13 is blocked, circulating eosinophils show reduced chemotaxis [31]. Thus, the drug inhibition of the IL4/13R axis and the effect on eosinophils migration to the tissues may be linked to our observation. A paper by Sanchez J. [32] considered that high levels of blood eosinophils and immunoglobulin E in asthmatic patients were correlated to the possibility of IgE autoantibody response to eosinophil proteins. There is empirical evidence indicating a correlation between these antibodies and the manifestation of severe asthma, suggesting their potential to initiate an inflammatory response. The ability to detect these autoantibodies may have significant implications for clinical practice, particularly with regard to the diagnosis and treatment of patients [32].

The trials LIBERTY NP SINUS-24 and LIBERTY NP SINUS-52 indicated that the levels of ECP in nasal lavage fluid were lowered at 24 weeks after the beginning of therapy with dupilumab [12]. Our results show a statistically significant increase for ECP and EOS is followed by a decrease below the baseline between three months and ten months.

It is important to note that over the course of the study duration, we documented neither serious adverse events (SAEs) nor asthma exacerbations. Furthermore, our findings confirm, in real life, the effectiveness of dupilumab in patients with CRSwNP in terms of quality of life (SNOT22) and in terms of FeNO and total IgE, in contrast with mepolizumab and benralizumab therapy that do not affect these two biomarkers [11,17]. In patients with asthma as a comorbidity, our findings confirm the effectiveness in terms of asthma control (ACT) and lung function, demonstrated by an increase in both the FEV1 and FEV1%, respectively.

IL-4 and IL-13 control the expression of inducible NO synthase, which generates NO [33]. FeNO and serum IgE levels decreased after dupilumab treatment, and a higher baseline FeNO level correlated with low FEV1, but it was also a predictor of a stronger response to dupilumab in terms of FEV1 [29], indicating that IL-4 and IL-13 are related to their levels. However, according to another study, the amount of ECP had a stronger correlation with reduced FEV1 than the eosinophil count did [34].

We did not find a correlation between nasal symptom improvement (SNOT22) and eosinophil count or ECP levels. The results of our investigation are in line with the findings of the SINUS 52 clinical trial, showing that dupilumab is effective in reducing the severity of symptoms associated with severe chronic rhinosinusitis with nasal polyps (CRSwNP) in patients who have either eosinophilic chronic rhinosinusitis (ECRS) or chronic rhinosinusitis that is not caused by eosinophils (non-ECRS). This study also shows that eosinophils are not a relevant biomarker for monitoring the efficacy of dupilumab in subjects who suffer from CRSwNP [35]. In agreement with this, our results from this prospective, real-life study, show that clinical effectiveness is also independent of ECP levels.

In contrast to anti-IL-5/5R mAbs approved for severe asthma, which showed a consistent and significant drop in ECP during treatment, we found a statistically significant increase in ECP after 3 months in 43.5% of subjects treated with dupilumab, which was followed by a decline towards baseline at 10 months in our cohort. Because of the limitations of the current study, more research with a larger sample size is required to replicate our findings and to investigate the possibility that the chosen biomarkers can serve as predictive indicators of efficacy.

## 5. Conclusions

The results of our study show that data collected in real life are comparable to those that were formerly documented in pivotal RCTs of dupilumab. Close to those studies, in addition to a significant improvement in clinical status, we also observed an increase in ECP and EOS during the first three months with dupilumab, which subsequently returned to normal levels. This occurred despite the fact that there was a significant improvement in clinical status. In terms of adverse events or lung function, greater ECP levels or EOS does not have a clinical impact. 

## Figures and Tables

**Figure 1 life-13-01884-f001:**
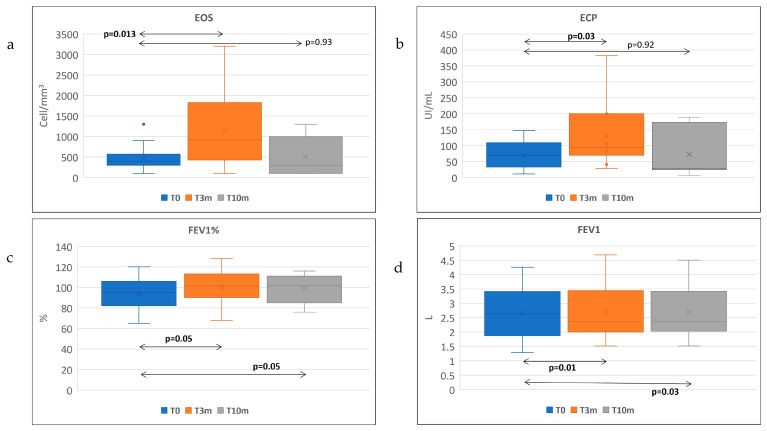
(**a**–**d**): Median value of EOS, ECP, FEV1 and FEV1% at each time point. EOS: eosinophil count (cell/mm^3^); ECP: Eosinophils Cationic Protein (U/mL). FEV1: Forced Expiratory Volume in the first second (L). For each variable, data are shown as media; the bars indicate the minimum and maximum measured values. Bold: significant *p*-value.

**Figure 2 life-13-01884-f002:**
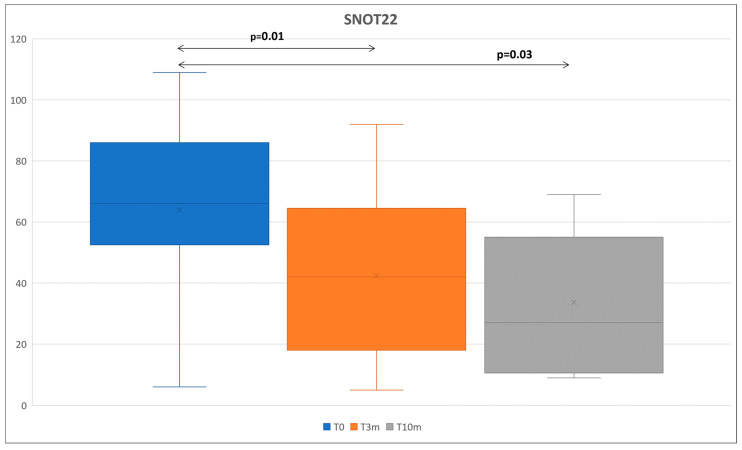
Median Values of SNOT22 at each time point. SNOT22: Sino-nasal outcome test 22; bars indicate the minimum and maximum measured values. Bold: significant *p*-value.

**Table 1 life-13-01884-t001:** Population characteristics in the population treated with dupilumab vs. control group.

Characteristic	Dupilumab (n = 21)	Control Group (n = 21)
**Age (years) Median ± IQR**	55 ± 17.3	45 ± 27.3
**Female, n**	11	10
**Never smoker (%)**	71	66
**CRSwNP (%)**	100	100
**Asthma (%)**	85	100
**Atopic**	9	11

IQR: interquartile range; CRSwNP: chronic rhinosinusitis with nasal polyps.

**Table 2 life-13-01884-t002:** Baseline ECP serum levels in the group treated with dupilumab and in the control group.

ECP	Minimum	Median	Maximum	IQR	*p* Value
**Control group**	4.5	69.5	147	65	0.35
**Dupilumab group**	20.3	120	318	128	

ECP: eosinophil cationic protein; IQR: interquartile range.

**Table 3 life-13-01884-t003:** Laboratory, questionnaires and clinical assessment at each time point compared with baseline in patients treated with dupilumab.

	T0 (Median ± IQR)	T3 (Median ± IQR)	*p* Value (T0–T3)	T0 (Median ± IQR)	T10 (Median ± IQR)	*p* Value (T0–T10)
**Eosinophil count (/μL)**	450 ± 233	860 ± 1079	*p* = 0.1	450 ± 275	300 ± 833	*p* = 0.93
**ECP (μg/L)**	69.5 ± 69	107 ± 118	*p* = 0.03	100 ± 65	36.5 ± 155	*p* = 0.90
**Total IgE (IU/mL)**	247 ± 333	99 ± 191	*p* = 0.002	195.5 ± 210	53.2 ± 78	*p* = 0.01
**FEV1 %**	96.5 ± 28	104 ± 9	*p* = 0.01	92 ± 19	104 ± 16	*p* = 0.03
**FEV1**	2.99 ± 1.4	2.34 ± 1.35	*p* = 0.52	2.23 ± 2.35	2.36	*p* = 0.05
**FeNO (ppb)**	48 ± 44	13 ± 18	*p* = 0.04	49 ± 27.5	46 ± 28	*p* = 0.01
**ACT**	21 ± 6	22.5 ± 7	*p* = 0.08	21 ± 11	22 ± 6	*p* = 0.09
**SNOT-22**	66.5 ± 34	42 ±46	*p* = 0.01	53 ± 45	27 ± 44	*p* = 0.02

ECP = eosinophilic cationic protein; FEV = Forced expiratory volume; FEV1 = Forced expiratory volume in 1 s; FeNo = Fractional exhaled nictric oxide; ACT = Asthma control test; SNOT-22 = Sino nasal outcome test.

## Data Availability

Data will be made available upon reasonable request to the corresponding authors.

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
