# Peer review of "Eosinophil Cationic Protein Variation in Patients with Asthma and CRSwNP Treated with Dupilumab"

_life, 2023, doi:10.3390/life13091884_

Round 1

Reviewer 1 Report

The manuscript presents not very innovative research. There are many studies on this topic. In general, the article is written clearly, although it does not bring much new data. I don't have many comments.

my comments:

1. The methodology is described very generally - please describe the laboratory methods used in detail - having only the data provided by the authors, it is not possible to find the research methods used

2. The results are described very generally, please add more details

3. References are not formatted in accordance with the requirements of the journal.

I don't see any major language problems other than minor stylistic errors.

Reviewer 2 Report

I appreciate the opportunity to review the manuscript for publication in MDPI Life. I feel that the topics are updated and interesting, and the manuscript is grossly well organized.

The authors assessed how the ECP levels and the peripheral blood eosinophil counts changed over time during the first course of dupilumab in CRSwNP asthmatic patients. I have a few comments as follows.

The abstract should include patient enrollment more in detail; eg. the number, proportion of BA and CRS status.

The authors should assess nasal polyp scores and sinus CT scores in the study setting. These are necessary to evaluate the efficacy of dupilumab.

A key article is missing in sentences in L80.

Tokunaga T, Sakashita M, Haruna T, Asaka D, Takeno S, Ikeda H, Nakayama T, Seki N, Ito S, Murata J, Sakuma Y, Yoshida N, Terada T, Morikura I, Sakaida H, Kondo K, Teraguchi K, Okano M, Otori N, Yoshikawa M, Hirakawa K, Haruna S, Himi T, Ikeda K, Ishitoya J, Iino Y, Kawata R, Kawauchi H, Kobayashi M, Yamasoba T, Miwa T, Urashima M, Tamari M, Noguchi E, Ninomiya T, Imoto Y, Morikawa T, Tomita K, Takabayashi T, Fujieda S. Novel scoring system and algorithm for classifying chronic rhinosinusitis: the JESREC Study. Allergy. 2015 Aug;70(8):995-1003. doi: 10.1111/all.12644. Epub 2015 May 26

In Figs 2 and 3, the plots should be error-bar styles to confirm inter-individual differences. The data on the control group should also be displayed.

“type 2 innate lymphoid cells” in L73 should be capitalized.

“controlled by a placebo” in L102: The reference is missing.

Please define “Atopic” in Table 1.

I have an argument about what authors intend to propose as for eosinophilia and dupilumab efficacy. The authors should cite and discuss the following key article.

Fujieda S, Matsune S, Takeno S, Ohta N, Asako M, Bachert C, Inoue T, Takahashi Y, Fujita H, Deniz Y, Rowe P, Ortiz B, Li Y, Mannent LP. Dupilumab efficacy in chronic rhinosinusitis with nasal polyps from SINUS-52 is unaffected by eosinophilic status. Allergy. 2021 May 16. doi: 10.1111/all.14906. Epub ahead of print. PMID: 33993501.

Round 2

Reviewer 1 Report

Dear Authors, I appreciate the corrections made, but in my opinion they are still insufficient.
Here are my other comments:
- lines 195-201: please supplement the methods with the manufacturers of these tests/analyzers
- lines 198-200: please provide information about the analyzers/systems on which these tests were
performed
- table 2: ECP units should be entered in the table, not in the table legend
- please format the list of references in accordance with the requirements of the journal

English is acceptable

Author Response

Dear editors and reviewers,

We modified the manuscript as detailed below, carefully taking into account the constructive and very useful comments provided.

POINT BY POINT REPLY

Thank you very much for your interesting comments and suggestions. Below we explain how we have incorporated them into this new version of the manuscript in order to improve the paper.

1)Dear Authors, I appreciate the corrections made, but in my opinion they are still insufficient.
Here are my other comments:
- lines 195-201: please supplement the methods with the manufacturers of these tests/analyzers

We thank the reviewer for his/her comment, we added the information according to your suggestions

2)  lines 198-200: please provide information about the analyzers/systems on which these tests were
performed

We thank you for your comment, we correct the manuscript accordingly

3) table 2: ECP units should be entered in the table, not in the table legend

We thank you for your comment, we agree, we correct the table 2 accordingly

4) please format the list of references in accordance with the requirements of the journal

We thank you for your comment, we fortmatted the references in accordance with the journal guidelines

Hoping you will find the manuscript of interest for your distinguished Journal

Yours sincerely

Davide Firinu

Reviewer 2 Report

I appreciate the opportunity to review again the manuscript for publication in MDPI Diagnostics. I reckon that the manuscript has been revised and improved in part in accordance with the reviewers’ comments. However, the amendments in the revised manuscript still fail to support the highlights of the article. No additional data have been presented in the following points.

L168: "We enrolled a total of 21 patients treated with dupilumab and 21 asthmatic patients not in biologicals that represent the control group (Table 1)."

The data on the control group should also be displayed in contrast with those of Figs 2 and 3.

The references should be listed in accordance with the Journal Instructions.

Author Response

Dear editors and reviewers,

We modified the manuscript as detailed below, carefully taking into account the constructive and very useful comments provided.

POINT BY POINT REPLY

I appreciate the opportunity to review again the manuscript for publication in MDPI Diagnostics. I reckon that the manuscript has been revised and improved in part in accordance with the reviewers’ comments. However, the amendments in the revised manuscript still fail to support the highlights of the article. No additional data have been presented in the following points.

1)L168: "We enrolled a total of 21 patients treated with dupilumab and 21 asthmatic patients not in biologicals that represent the control group (Table 1)." The data on the control group should also be displayed in contrast with those of Figs 2 and 3.

We thank the Reviewer for the comment. However, according to the aim of our study, we would highlight that we evaluated the ECP in the control group only at the baseline (and the medians and distribution are reported in the Table 2). We think that a boxplot is not very useful in this case. If the Reviewer still would suggest that a graphical plot of data is required, we are available to replace the Table 2.

2)The references should be listed in accordance with the Journal Instructions.

We thank you for your comment and we agree, we formatted the references accordingly. Please note that we used the “free format submission” option that is available for “Life” and a software  for reference management. Thus, some small details in bibliography are going to be resolved and double checked at a later stage.

Hoping you will find the manuscript of interest for your distinguished Journal

Yours sincerely

Davide Firinu